# Large-Scale Application of Double-Stranded RNA Shows Potential for Reduction of Sacbrood Virus Disease in *Apis cerana* Apiaries

**DOI:** 10.3390/v15040897

**Published:** 2023-03-31

**Authors:** Mi-Sun Yoo, A-Tai Truong, Hana Jeong, Do-Hyun Hahn, Ju-Seong Lee, Soon-Seek Yoon, So-Youn Youn, Yun-Sang Cho

**Affiliations:** 1Parasitic and Honey Bee Disease Laboratory, Bacterial and Parasitic Disease Division, Department of Animal and Plant Health Research, Animal and Plant Quarantine Agency, Gimcheon 39660, Gyeongsangbuk-do, Republic of Korea; 2Faculty of Biotechnology, Thai Nguyen University of Sciences, Thai Nguyen 250000, Vietnam

**Keywords:** sacbrood virus, RNA interference, double-stranded RNA, honey bee, *Apis cerana*

## Abstract

Sacbrood virus (SBV) infection has emerged as a remarkable threat to *Apis cerana* colonies in South Korea, necessitating prompt control measures. In this study, RNA interference (RNAi) targeting the VP3 gene was developed to assess its safety and efficacy in protecting and treating SBV in vitro and in infected colonies in South Korean apiaries. The efficacy of VP3 double-stranded RNA (dsRNA) was demonstrated in laboratory-based experiments, wherein infected larvae treated with VP3 dsRNA exhibited a 32.7% increase in survival rate compared to untreated larvae. Data from a large-scale field trial indicate the efficacy of dsRNA treatment since none of the treated colonies had symptomatic SBV infections, whereas disease was observed in 43% (3/7) of the control colonies. In the 102 colonies exhibiting symptoms of SBV disease, RNAi treatment provided partial protection with weekly treatment, prolonging the survival period of colonies to 8 months compared to 2 months in colonies treated at 2- and 4-week intervals. Therefore, this study demonstrated that RNAi is a valuable tool for preventing SBV disease outbreaks in healthy and low-level SBV-infected colonies.

## 1. Introduction

Sacbrood virus (SBV) is a positive-sense, single-stranded RNA virus that belongs to the family Iflaviridae. The genome of SBV is 8832 nucleotides (nt) long and encodes for a single polyprotein (2858 aa long) that is cleaved into nonstructural proteins and capsid proteins, including VP1, VP2, VP3, and VP4 [1]. SBV was first discovered in the western honey bee, *Apis mellifera*. The infection results in larval death, ecdysial fluid, and sac formation [2,3,4]. The infection of SBV in adult bees is not easily discernible [5], but shortened lifespans and reduced pollen collection have been reported in infected adult bees [6,7]. Since its detection in the eastern honey bee, *Apis cerana*, SBV has become a devastating pathogen and the primary cause of colony collapse in this honey bee species in Vietnam, China, and the Republic of Korea (ROK) [8,9,10,11,12]. Different SBV genotypes have been identified in *A. cerana* in these countries, with a mutation in the minor capsid protein gene located between the VP1 and VP3 genes [13,14]. SBV has now become a prevalent pathogen in honey bees and has had a considerable adverse effect on the beekeeping industry.

Various methods for SBV treatment have been tested, such as queen replacement, nutritional supplements, antibody treatment, and herbal medicines [7]. However, none of these methods have been widely adopted due to their shortcomings. For instance, the use of the herbal extract thymol to inactivate SBV in *A. cerana* resulted in decreased larval survival rates in vitro due to its side effects [15]. In ROK, silver ions were used for the treatment and prevention of SBV in *A. cerana* [8], but they were not commercially viable because of food safety concerns. Although lactic acid bacteria possess immunity-enhancing properties, their application can trigger hygienic behavior among nurse bees, leading to the elimination of healthy larvae that have received the bacteria [16].

The ability of insect cells to take up environmental and artificial double-stranded RNA (dsRNA) via scavenger receptor-mediated endocytosis has been demonstrated [17,18,19]. When present in the cytoplasm, dsRNA activates the small interfering RNA (siRNA) pathway, which involves the cleavage of dsRNA into small fragments of 18 nt–24 nt by the RNase type III enzyme Dicer-2 [20]. The cleavage is then carried out by Argonaute-2, a component of the RNA-induced silencing complex (RISC) [18,21,22]. dsRNA also triggers the non-sequence specific antiviral pathway in honey bees, which stimulates the non-specific immune response to limit viral infections [23,24]. RNA interference (RNAi) has been reported to inhibit and control honey bee parasites, such as injecting dsRNA of the V-ATPase subunit A into small hive beetle (*Aethina tumida* Murray) larvae, resulting in 100% mortality [25] and feeding honey bees dsRNA of *Varroa destructor* genes, which can decrease the mite population by over 60% [26]. RNAi is also a promising method for preventing honey bee viral pathogens and could be developed for field application [10,27,28,29,30].

Administration of RNAi specific to SBV in *A. cerana* was shown to be effective in blocking viral propagation and increasing the survival rate of larvae in vitro, suggesting that it could be a promising method for controlling SBV in *A. cerana* [10,31]. However, no field studies on the efficacy of RNAi for wide application in SBV treatment and prevention have been reported. Accordingly, this study was conducted to analyze the efficacy of RNAi for the prevention and treatment of South Korean SBV in *A. cerana* and to develop a procedure for its application in apiaries.

## 2. Materials and Methods

### 2.1. Isolation of Sacbrood Virus

Confirmation of the presence of SBV in honey bee samples was achieved through viral isolation and real-time reverse transcription polymerase chain reaction (RT-qPCR) [32,33]. For the isolation of SBV, the 5th instar larvae were pulverized using a mortar and pestle. The samples were filtered using 0.45 µm and 0.2 µm syringe filters. After sonicating for 30 s (2 s on and 3 s off), the solution was used for SBV purification via sucrose density gradient centrifugation (10–50% range) at 32,500 rpm for 4 h using an SW 41 Ti Swinging-Bucket Rotor (Beckman Coulter Inc., Brea, CA, USA). The SBV band corresponding to the 40% sucrose gradient was harvested and then centrifuged at 25,000 rpm for 12 h at 4 °C. The resulting SBV was resuspended in distilled water (ddH_2_O). Photographs were taken with a transmission electron microscope (H-7100FA, Hitachi Ltd., Tokyo, Japan) to confirm the viral particles of 27.8 ± 0.4 nm in size, similar in shape to other picornaviruses (Appendix A).

### 2.2. Extraction of Viral RNA

RNA was extracted from five larvae from the 3rd to 5th instar larval samples that were collected in 1 mL phosphate-buffered saline (PBS) using a tissue grinding tube with 2.381 mm steel beads (SNC, Hanam, ROK) and 100 µL of the homogenate. For adult bees, two bees were placed in a tissue grinding tube with 2.381 mm steel beads (SNC) and then homogenized with a Precellys 24 Tissue Homogenizer (Bertin Instruments, Montigny-le-Bretonneux, France) after adding 1 mL of PBS solution. Two hundred µL of homogenate were used for RNA extraction. The RNA of SBV was extracted using the QIAamp Viral RNA Mini Kit (QIAGEN, Hilden, Germany) following the manufacturer’s instructions.

### 2.3. Construction of SBV Recombinant Plasmid

The procedure for synthesizing SBV cDNA from extracted RNA entailed utilizing the SuperScript III First-Strand Synthesis System for RT-PCR (Thermo Fisher Scientific, Waltham, MA, USA). The manufacturer’s instructions were followed for cDNA synthesis. In brief, a reaction mixture of 10 µL, containing 4 µL of RNA, 5 µL of reverse primer (10 pmol/µL; T7VP3R), and 1 µL of a 10 mM dNTP mix, was incubated at 65 °C for 5 min and then placed on ice for 1 min. Then, a solution containing 2 µL of 10× RT buffer, 4 µL of 25 mM MgCl_2_, 1 µL of RNase OUT, 1 µL of SuperScript III RT, and 2 µL of ddH_2_O was added to obtain a reaction mixture of 20 µL. Reverse transcription was conducted by incubating the reaction mix at 50 °C for 50 min, then at 85 °C for 5 min, and finally, after chilling on ice, 1 µL RNase H was added, and the mix was incubated at 37 °C for 20 min. The cDNA was then used for PCR with the primer set T7VP3F: 5′-taa tac gac tca cta tag ggc gaA GAT GTG AAC GCT TAC CCT GAT-3′/T7VP3R: 5′-taa tac gac tca cta tag ggc gaC TCC TCG CAT ATA CAC CAA AAC TT-3′ for VP3 fragments [10], and TOPsimple DryMIX-HOT PCR premix (Enzynomics, Daejeon, ROK). The 20 µL PCR mixture consisted of 1 µL of cDNA, 1 µL (10 pmol) of each primer, and 17 µL of ddH_2_O. The PCR conditions involved an initial denaturation at 94 °C for 5 min, followed by 40 cycles at 94 °C for 15 s, 62 °C for 15 s, 72 °C for 15 s, and the final extension at 72 °C for 10 min. The DNA fragment was confirmed for the presence of the VP3 band (596 bp), extracted, and purified using the QIAquick Gel Extraction Kit (QIAGEN). The TOPcloner TA Kit (Enzynomics) was used to construct a recombinant plasmid carrying the VP3 fragment, which was transformed into competent Escherichia coli DH5α (Enzynomics). The recombinant plasmid was sequenced by GenoTech (Daejeon, ROK) and was compared to the SBV sequences reported in NCBI using the Basic Local Alignment Search Tool (BLAST). The recombinant plasmid showed 99% similarity with the South Korean SBV (NCBI accession no.: HQ322114; Appendix A). Finally, the recombinant plasmid was used as a DNA template for PCR to produce dsRNA.

### 2.4. Production of Anti-SBV dsRNA

The 596 bp dsRNA corresponding to VP3 of the South Korean SBV (NCBI accession no.: HQ322114; 1596–2191) was synthesized by transcribing the T7 promoter designed downstream of the T7VP3F/R primer pair sequence. We performed PCR using the primer pair, then used the PCR product as a template for producing VP3 dsDNA, and the in vitro Transcription T7 kit (for siRNA synthesis; No. 6140, TaKaRa Bio Inc., Kusatsu, Japan) was used. The dsRNA synthesis was performed according to the manufacturer’s instructions. Briefly, we mixed 2.5 µL of each NTP (50 nmol/µL), 3 µL of 10× reaction buffer, 1 µg DNA template, and 2 µL of T7 Enzyme Mix (TaKaRa). We then adjusted the mixture to 30 µL with nuclease-free ddH_2_O before incubating it at 42 °C for 2 h. Subsequently, we added 30 µL of LiCl solution (7.5 M) and incubated the mixture at −20 °C for 30 min to precipitate the RNA. We centrifuged the mixture at 16,000× *g* for 15 min and discarded the supernatant. Then, we added 1 mL of 70% ethanol and centrifuged the mixture again at 10,000× *g* for 5 min. After discarding the supernatant, we dissolved the RNA in 30 µL of elution buffer and then stored it at −70 °C until further use. A large quantity of dsRNA (Appendix A) was produced in collaboration with Genolution, Inc. (Seoul, ROK) to extend the application to apiaries in the field.

### 2.5. Real-Time Reverse Transcription PCR for Evaluation of SBV Inhibition by dsRNA

RT-qPCR was used to quantitatively evaluate the efficacy of dsRNA for inhibiting SBV in infected honey bees. We designed a specific primer pair, KSBV-PCR-F (5′-GACCAAGAAGGGAATCAG-3′) and KSBV-PCR-R (5′-CATCTTCTTTAGCACCAGTATCCA-3′), to amplify a 123 bp fragment of South Korean SBV (NCBI accession no.: HQ322114; 2306–2428; Appendix A). We performed PCR using the One Step TB Green PrimeScript RT-PCR Kit II (TaKaRa). The corresponding 20 µL reaction mix was composed of 2 µL of sample RNA, 1 µL (10 pmol) of each primer, 1 µL of PrimeScript One Step Enzyme Mix (TaKaRa), 10 µL of 2× One Step RT-PCR buffer (TaKaRa), and 5 µL ddH_2_O. We used the CFX96 Touch Real-time PCR Detection System (Bio-Rad, Hercules, CA, USA) for RT-qPCR. We performed reverse transcription at 42 °C for 30 min, followed by PCR at 95 °C for 5 min, then 40 cycles at 95 °C for 30 s, 55 °C for 30 s, and 72 °C for 30 s. To confirm positive results, the melting curve peaks were analyzed. To estimate the number of DNA copies, standard curves representing the relationship between the cycle threshold (Ct) of amplification and the initial number of DNA copies, ranging from 10^10^ to 10^1^ (10-fold dilution), was established in triplicate PCRs (Appendix A). The DNA copy number of standard DNA was calculated using the formula presented on the website (http://cels.uri.edu/gsc/cndna.html, accessed on 1 September 2018). The data regarding SBV quantification presented in this study were obtained from the final report of the project on SBV prevention using dsRNA in 2018. We could not access the raw data of the Ct value in RT-qPCR detection.

### 2.6. Efficacy of dsRNA In Vitro

The safety and efficacy of dsRNA were evaluated in vitro using artificially reared honey bee larvae. The larvae were transferred to the wells of 24-well plates and incubated at 35 °C and 80% humidity. Each larva received a daily dose of an artificial nutrient solution consisting of 6% D-glucose, 6% D-fructose, 1% yeast extract, 33% Gibco Grace’s insect medium (Thermo Fisher Scientific), and 50% royal jelly at a dose of 100 µL/day.

To determine the efficacy of dsRNA in inhibiting SBV in vitro, we evaluated the effects of dsRNA treatment on SBV infection in three-day-old larvae. SBV used for infection was obtained from naturally infected larvae that exhibited SBV-associated symptoms, including color and sac-like appearance due to fluid accumulation in larvae [2]. The presence of SBV was confirmed via RT-qPCR [32] and viral purification, as described previously. The liquefied sample was centrifuged at 13,000× *g* for 1 min to eliminate honey bee debris and other materials in the homogenate. The supernatant was filtered using 0.45 µm and 0.2 µm syringe filters. Finally, RT-qPCR was performed again to confirm the presence of SBV in the filtered solution and to calculate the quantity of SBV in the solution. The solution was then used to prepare a feeding solution by adding other nutrients in ratios similar to those described above. The feeding solution, which contained 10^8^ copies of SBV, was administered in a single dose to each larva being reared. The reared larvae and feeding solution used in the evaluation were confirmed to be free of other viral pathogens, including deformed wing virus (DWV), acute bee paralysis virus (ABPV), black queen cell virus (BQCV), chronic bee paralysis virus (CBPV), Israeli acute paralysis virus (IAPV), and Kashmir bee virus (KBV), using a LiliF™ SBV/KSBV/DWV/BQCV RT-qPCR Kit and LiliF™ ABPV/KBV/IAPV/CBPV RT-qPCR kits (iNtRON Biotechnology, Inc., Seongnam, ROK) [32]. The experiment was conducted with six groups of larvae. Group 1 contained larvae in four 24-well plates (*n* = 96) that were orally administered with dsRNA 1 day after SBV infection. The dsRNA solution was prepared by mixing 1 µg of dsRNA with the feeding solution, which was then supplied to each larva in one dose. Group 2, serving as the control, included SBV-infected larvae in four plates that were not fed dsRNA. In Group 3, the larvae were fed only dsRNA in the feeding solution (no SBV infection) to evaluate the safety of administering dsRNA to larvae in vitro. Group 4 received only feeding solutions (no dsRNA or SBV). Healthy larvae in Group 5 were administered green fluorescent protein (GFP) RNA, as were those in the SBV infection group (Group 6). The experiment for each group was repeated three times within 8 days of each trial. The number of living larvae was counted daily until the 8^th^ day of the inoculation period to calculate the survival rate of larvae.

### 2.7. Field Efficacy

The effectiveness of dsRNA in preventing SBV was first evaluated in 78 colonies across 17 apiaries, where 18 colonies were initially treated with 10 mg of dsRNA/hive via spray or oral administration. For the spray method, the hives were opened, and a 5 mL feeding solution containing 10 mg dsRNA was sprayed on both sides of each honeycomb, whereas the oral administration was achieved through the use of a feeding pouch containing an equal volume of dsRNA-feeding solution. Afterward, dsRNA application was extended to 60 other colonies at 20 mg/hive via oral administration. Before administering dsRNA, the presence of SBV in the colonies was confirmed by RT-qPCR. However, no symptoms of SBV disease were observed in any of the colonies. The dsRNA was applied five times to the hives, with a one-week interval, during the summer months of May to June. Additionally, seven colonies from four other apiaries were not treated with dsRNA and were designated as the control group. The apiaries were located in close proximity (approximately 6 km) to the disease outbreak regions.

Finally, dsRNA was applied on a larger scale to 269 colonies from 33 different apiaries in ROK, of which 167 colonies were asymptomatic and used to evaluate SBV prevention, while 102 colonies with SBV symptoms were used to assess the efficacy of dsRNA in disease treatment. Each oral administration involved a dose of 20 mg dsRNA/hive. The large quantity of dsRNA was produced by Genolution, Inc. (Seoul, ROK) at a cost of 10 USD for each 20 mg dsRNA. In the colonies, the queens were not confined during the dsRNA application period.

### 2.8. Data Analysis

The comparison was performed to verify the significant difference in the survival rate of larvae among the groups in vitro test, and the one-way analysis of variance was performed, followed by Tukey’s pairwise comparison test to determine the significance between groups [34]. The efficacy of dsRNA in inhibiting SBV in infected colonies in the field was evaluated by comparing SBV cDNA copies before and after treatment. The Mann-Whitney U test was employed to establish a significant difference. All statistical analyses were performed using the Paleontological statistics software package for education and data analysis software (PAST v. 4.0) [35]. A probability level of less than 5% was considered to indicate statistical significance.

## 3. Results

### 3.1. Efficacy of dsRNA In Vitro

The efficacy of dsRNA in inhibiting SBV was demonstrated in the survival rate of larvae reared in vitro. A significant difference (*p* = 0.004) was observed in the survival rate of SBV-infected larvae between the group that received dsRNA and the group that did not. The survival of SBV-infected larvae without dsRNA treatment decreased rapidly to 48.8% on the 8th day of inoculation (SBV group; Figure 1). In contrast, the infected larvae that received the dsRNA treatment exhibited a higher rate of survival, with a survival rate of 81.5% at the end of the experimental period (SBV dsRNA + SBV group; Figure 1). Furthermore, oral administration of dsRNA was safe for larvae, as demonstrated by the survival rate (92.3%) of the group that was fed only dsRNA (SBV dsRNA group; Figure 1), which did not significantly differ from that (94.6%) of the larvae that received the feeding solution without SBV (*p* = 0.999; No treatment group; Figure 1). The survival rates of larvae in the groups that received GFP RNA were 93.9% and 66.1% for the SBV-free group (GFP dsRNA group) and the SBV-infected group (GFP dsRNA + SBV group; Figure 1), respectively. Although the SBV-infected larvae that received GFP dsRNA (GFP dsRNA + SBV group; Figure 1) exhibited a higher survival rate (66.1%) than that of the infected larvae without dsRNA treatment (48.8%; SBV group; Figure 1), no significant difference was observed between the two groups (*p* = 0.182).

### 3.2. Field Application of dsRNA for the Prevention and Treatment of SBV

The prophylactic administration of dsRNA resulted in the absence of SBV disease symptoms in all 78 *A. cerana* colonies that received dsRNA via spray or oral administration. All colonies that received doses of 10 mg or 20 mg of dsRNA demonstrated healthy conditions (Table 1; Figure 2a). Conversely, symptoms of SBV disease were observed in three out of the seven control colonies (42%), which were not administered dsRNA (Figure 2b). Two weeks after the last dsRNA treatment, all colonies in the treatment groups survived in a healthy condition, whereas the three colonies with SBV disease symptoms were lost.

The quantitative detection of SBV in 22 out of the 78 colonies randomly selected from six apiaries showed a significant decrease in SBV DNA per bee after four weeks of dsRNA treatment (*p* = 0.0001). SBV relative cDNA per bee in the dsRNA-treated colonies ranged from 2.5 × 10^1^ to 2.26 × 10^5^ copies (Figure 3 and Appendix A). Notably, none of the 78 colonies showed any symptoms of SBV disease following dsRNA treatment (Figure 2a). In contrast, the control group without dsRNA treatment showed increased SBV cDNA levels in two out of seven colonies, with SBV cDNA levels ranging from 1.13 × 10^7^ to 1.34 × 10^8^ copies/bee in one colony and 3.8 × 10^11^ copies/bee in the other colony, which collapsed one week after SBV detection (Figure 3 and Appendix A).

As for the large-scale dsRNA application, we used 167 colonies that tested positive for SBV by RT-qPCR but had no visible symptoms of the disease as the prevention group, with application intervals of 2 or 4 weeks (Table 2). All of the colonies maintained healthy conditions throughout the inspection period. In the group with symptoms of SBV disease (102 colonies), the administration of dsRNA showed partial effectiveness in colonies treated at a 1-week interval. The treated colonies survived for up to 8 months. However, the colonies with application intervals of two and four weeks showed no effectiveness, and they disappeared after 2 months (Table 2).

## 4. Discussion

The dsRNA designed for use against SBV was produced and tested on in vitro-reared larvae as well as for large-scale field application. Previous studies have demonstrated the efficacy of targeting the VP1 gene for SBV prevention, as reported by Liu et al. (2010) [10]. However, recent findings identified the target gene as the capsid gene VP3 [13]. Therefore, this study targeted the VP3 gene for RNA interference (RNAi) against SBV. The efficacy and safety of dsRNA in preventing and treating SBV were demonstrated [28].

Viral diseases are a major cause of honey bee colony losses globally, with more than 30 viruses identified [36,37]. However, no specific chemotherapy exists for the prevention and treatment of these viral diseases. Traditional methods for the management of viral diseases are good practices for beekeeping and help with the early diagnosis of diseases and the selection of honey bee strains that are resistant to viruses [38]. RNAi effectively inhibited IAPV and SBV [10,30], leading to the development of dsRNA field treatment for IAPV in 2010 [29]. However, the high costs of dsRNA production have hindered large-scale production. In vivo dsRNA production methods were introduced in 2016 [31] to decrease production costs compared to the in vitro method [31]. The in vivo method estimated a cost of 224 USD per mg dsRNA [31], which is still high for field application, which requires approximately 20 mg for one dose. Reducing the cost of dsRNA production is therefore pertinent for widespread RNAi application.

Although SBV causes the death of infected larvae [2], its symptoms in infected adult honey bees remain unclear [5,39]. Therefore, during the field application of dsRNA for disease prevention in infected colonies, the queens were confined for the first 3 weeks. This mitigated the replication of SBV and was helpful for dsRNA treatment in the infected colonies. After five dsRNA administrations, SBV levels decreased below the threshold required to generate the disease in infected colonies (≤2.26 × 10^5^ copies/bee) [40]. Further investigation is required to determine the duration of dsRNA administration required to maintain SBV at safe levels and the time required for SBV to reach the replication level that causes the associated disease after terminating the use of dsRNA.

Recent research has demonstrated the digestion and transmission of double-stranded RNA (dsRNA) among bees in a single colony via hemolymph and worker and royal jellies [41]. It has also been shown that dsRNA can be transmitted to parasites that feed on honey bee hemolymph [26]. In order to reduce IAPV, honey bees were fed with dsRNA for field application of RNA interference (RNAi) [29]. In this study, we initially supplied dsRNA to the honey bee colonies by both feeding and spraying. Although the spraying method can rapidly deliver the dsRNA solution to a large number of bees, allowing for a shorter exposure time, it requires more handling time and results in a significant amount of solution being wasted as it is attached to both the bees and the comb without being consumed. Moreover, the efficacy of SBV prevention was found to be similar for both methods. Therefore, for large-scale field applications, the feeding method was ultimately chosen.

Studies have shown that honey bee colonies with natural infections of single-strand RNA viruses can set out siRNA responses against viruses [42]. During viral replication, the dsRNA molecules of the positive- and negative-sense ssRNA viruses trigger the endonuclease, Dicer, to cleave the long dsRNA into 21–23 base siRNAs. These siRNAs induce RISC to limit viral replication [43]. The high prevalence of SBV in Korean apiculture could also result in the effectiveness of a natural RNAi response in the *A. cerana* apiaries. Therefore, the high efficacy of dsRNA in protecting colonies from SBV disease was demonstrated in this study. Furthermore, the study demonstrated an increase in gene expression involved in RNAi, Toll, lmd, and JAK-STAT pathways when honey bees were treated with dsRNA. Notably, dsRNA triggers a non-sequence-specific antiviral pathway in honey bees [23,24]. Therefore, the survival rate of SBV-infected larvae receiving non-sequence specific dsRNA (GFP dsRNA) was higher than that of the SBV-infected group without dsRNA treatment.

## 5. Conclusions

The application of dsRNA was safe for honey bees, and the usefulness of dsRNA for preventing SBV disease outbreaks, as well as for the treatment of SBV infection, was demonstrated. The result initially indicates that dsRNA can be used as a specific drug for the prevention and treatment of SBV disease in beekeeping. However, further investigation is pertinent to validate these findings by expanding in vitro evaluation and assessing the replication in the negative control colonies in the large scale of dsRNA application. Additionally, it is crucial to determine the duration of protection after each administration, the adequate administration frequency, and the optimal season for application.

## Figures and Tables

**Figure 1 viruses-15-00897-f001:**
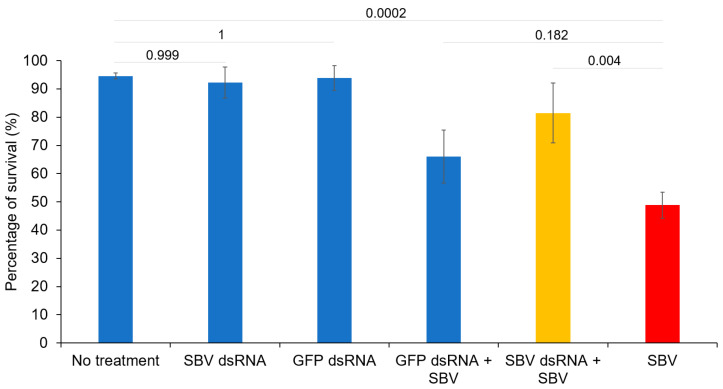
Treatment of sacbrood virus (SBV) using VP3 double-stranded RNA (dsRNA) in honey bee larvae reared in vitro. The survival rate of larvae in each group was calculated on the eighth day of the inoculation period based on three replicated trials. dsRNA of SBV VP3 and GFP was supplied to the healthy larvae in the “SBV dsRNA” and “GFP dsRNA” groups, respectively, to assess the safety of dsRNA to the larvae. dsRNA of each gene was also supplied to SBV-infected larvae to test the efficacy of SBV treatment (in the GFP dsRNA + SBV and SBV dsRNA + SBV groups). One SBV-infected group (SBV) and one healthy group did not receive dsRNA treatment (No treatment). Tukey’s pairwise comparison *p* values between the groups are shown.

**Figure 2 viruses-15-00897-f002:**
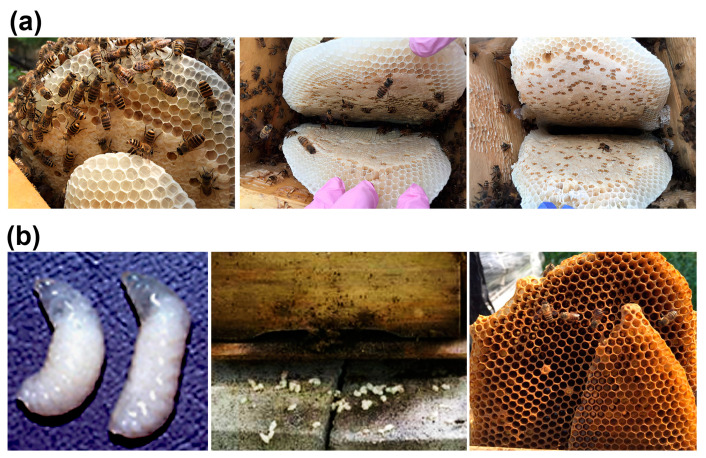
Colonies of *Apis cerana* after sacbrood virus (SBV) infection and double-stranded RNA (dsRNA) treatment. The SBV-infected colonies were treated with dsRNA five times at 1-week intervals. The treatment resulted in an increase in colony size, with larvae developing into pupae within capping cells and no visible symptoms of the disease within the hives (**a**). Conversely, colonies that did not receive dsRNA treatment exhibited symptoms of SBV disease, including dead larvae with ecdysial fluid visible in the comb, which was removed from the hive by adult nurse bees, and a reduction in the number of adult bees (**b**).

**Figure 3 viruses-15-00897-f003:**
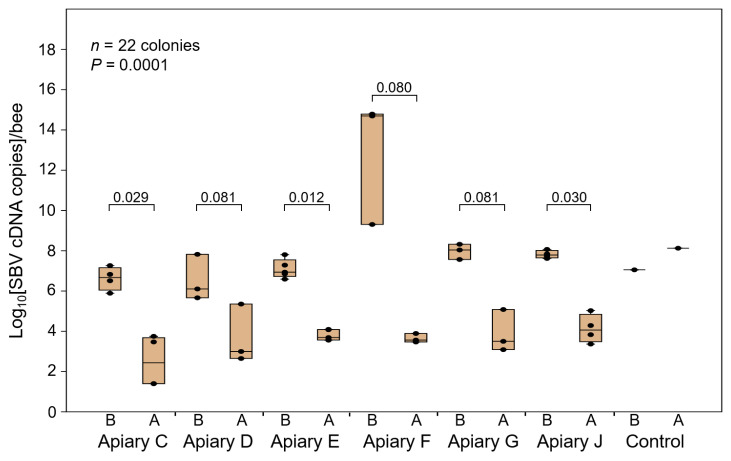
Relative SBV levels in double-stranded RNA (dsRNA) treatment apiaries. A total of 22 colonies from six randomly selected apiaries (apiaries C, D, E, F, G, and J) out of the 78 dsRNA-treated colonies belonging to 17 apiaries were quantitatively assessed for SBV levels before and after the 5th dsRNA administration. Additionally, SBV detection was performed on one of the seven control colonies without dsRNA treatment, which showed an increase in SBV. Significant differences in SBV DNA in the colonies were observed (*n* = 22, *p* = 0.0001) before and after treatment. The *p*-value for the comparison before and after treatment in each apiary is shown. Only one colony in the control group was measured. Therefore, the statistical analysis was not performed for this group. “A” and “B” denote the same colonies in each apiary after and before dsRNA administration, respectively. qPCR data represent estimated abundance based on equal volumes of sample lysate, which may not accurately reflect the actual SBV copy number per sample or per bee.

**Table 1 viruses-15-00897-t001:** Field application of double-stranded RNA (dsRNA) for the prevention of sacbrood virus (SBV).

Groups	Application Routes (Concentration)	Apiary No.	Colony No.	No. of Colonies with SBV Symptoms (%)	SBV Presence ^1^	Surviving Colonies ^2^
dsRNA treatment	Spray (10 mg)	5	9	0 (0)	+	9
	Oral (10 mg)	5	9	0 (0)	+	9
	Oral (20 mg)	7	60	0 (0)	+	60
	Subtotal	17	78	0 (0)	+	78
Negative control	No treatment	4	7	3 (42)	+	4

^1^ Positive results for SBV detection were acquired via RT-qPCR. ^2^ The number of surviving colonies was observed during the 2-month dsRNA treatment period.

**Table 2 viruses-15-00897-t002:** Extended field application of double-stranded RNA (dsRNA) for the prevention and treatment of sacbrood virus (SBV).

Groups	Before Administration	Apiaries (Hives)	Administration Interval (Week(s))	After Administration	Results	Efficacy
Prevention	Infected; no disease symptoms	1 (33)	4	Healthy	Healthy and split	Effective
		16 (134)	2	Healthy	Healthy and split	Effective
Treatment	Disease symptoms	4 (18)	4	Pulled out larvae	Colony loss within 2 months (100%)	Not effective
		11 (78)	2	Pulled out larvae	Survival for more than 2 months but finally loss (100%)	Not effective
		1 (6)	1	Less pulled out larvae	Keeping colony until 8 months; finally loss (100%)	Partially effective
	Total	33 (269)				

## Data Availability

The data presented in this study are available in this article and Appendix A.

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
