# Peer review of "Large-Scale Application of Double-Stranded RNA Shows Potential for Reduction of Sacbrood Virus Disease in Apis cerana Apiaries"

_viruses, 2023, doi:10.3390/v15040897_

Round 1

Reviewer 1 Report

The manuscript presented by Mi-Sun Yoo et al., provided the helpful therapy to SBV disease. There are plenty of work to RNAi in apiaries for SBV prevention and very labor-intensive. And the authors also mentioned the limitation due to the cost of dsRNA, so I recommend to accept the manuscript, but there have been a few issues through out the manuscript that I hope authors to answer them:

(1) about RNAi research of SBV, VP1,VP2, RdRp were served as target for RNAi before, in your work, dsRNA for VP3 was acquired very good results to inhibit SBV. So could you give any comments in disscussion about RNAi efficacy? Besides, did you detect the expression of VP3 after RNAi? Did VP3 was down-regulated?

(2) In section of materials and methods, do you set the does of dsRNA for sprayed in honey comb,or feeding? I mean how did you known the hives were administered with 20mg dsRNA five times with a 1-week interval for one hive was effective?

(3)in the figue1, Is there any significant difference between the RNAi+SBV group and the SBV group?

(4)in my opinion, both Figures 3 and 4 illustrate that the amount of SBV is reduced after dsRNA treatment, fig4 could put into supplements.

(5)there was confused about the quantification of SBV, in the manuscript,  KSBV-PCR-F (5′-GACCAAGAAGGGAATCAG-3′) and KSBV-PCR-123R (5′-CATCTTCTTTAGCACCAGTATCCA-3′) was used, if so, the primers in table1 were used for what? In addition, KSBV-PCR-123R should revise to KSBV-PCR-R.

(6)the text should be revised with attention because there are a lot of grammatical errors and typos.

Line 129, Table2, T7VP1F, target gene was VP3, could you explain why T7VP1 target VP3?

Line 161,for 1 m to eliminate should revise to for 1 min to eliminate;

Author Response

Reviewer 1

The manuscript presented by Mi-Sun Yoo et al., provided the helpful therapy to SBV disease. There are plenty of work to RNAi in apiaries for SBV prevention and very labor-intensive. And the authors also mentioned the limitation due to the cost of dsRNA, so I recommend to accept the manuscript, but there have been a few issues throughout the manuscript that I hope authors to answer them:

Answer: Thank you for your comments and suggestions, we revised the manuscript according to each suggestion below:

  • about RNAi research of SBV, VP1, VP2, RdRp were served as target for RNAi before, in your work, dsRNA for VP3 was acquired very good results to inhibit SBV. So could you give any comments in disscussion about RNAi efficacy? Besides, did you detect the expression of VP3 after RNAi? Did VP3 was down-regulated?

Answer: The target gene selected for RNAi in this study located at the same position as in the previously study by Liu et al. (2010) (Liu, X.; Zhang, Y.; Yan, X.; Han, R. Prevention of Chinese sacbrood virus infection in Apis cerana using RNA interference. Curr. Microbiol. 2010, 61, 422–428, doi:10.1007/s00284-010-9633-2), that was named as VP1. However, the recently published paper by Procházková et al., (2018) (Procházková, M.; Füzik, T.; Škubník, K.; Moravcová, J.; Ubiparip, Z.; PÅ™idal, A.; Plevka, P. Virion structure and genome delivery mechanism of sacbrood honeybee virus. Proc. Natl. Acad. Sci. U. S. A. 2018, 115, 7759-7764, doi:10.1073/pnas.1722018115) the selected gene was identified as VP3. Therefore, the name of the target gene was used as VP3 in this study. The information is added in the discussion section between Line 315-318. In this study the VP3 gene was inserted only in the pTOP TA V2 vector (Enzynomics; Line 119-120), after sequencing identification the recombinant plasmid was used as DNA template for PCR, then the PCR product was used for production of dsRNA. We did not have experiment for expression of the gene.

  • In section of materials and methods, do you set the does of dsRNA for sprayed in honey comb, or feeding? I mean how did you known the hives were administered with 20mg dsRNA five times with a 1-week interval for one hive was effective?

Answer: We supplied dsRNA to the hive with 10 mg and 20 mg by two methods, spraying and feeding to figure out the efficient method for the application. Finally, we found that the result of supplying 10 mg and 20 mg was not different, the two different amount both showed the high efficiency of SBV prevention. And the cost for each application with 20 mg dsRNA was just around 10USD. Therefore, we finally used 20 mg for each use in the large application to make sure that the amount of RNA remains enough in the hive during the period of application. The finally efficient interval for the application of dsRNA was set based on our initial result that after supplying dsRNA 7 days (in the figure below, this result was not included in the manuscript due to the small number of colonies used for the evaluation), and the efficiency of one-week interval was higher than 2 weeks in the colonies with symptom of SBV disease (result in Table 3; Line 310-311).

Figure. Prevention of sacbrood virus (SBV) using VP3 double-stranded RNA (dsRNA) in the control colonies. The experiment was conducted in three different colonies; RT-qPCR was used to evaluate the efficiency of dsRNA against SBV via the quantitative detection of SBV (i.e. DNA copies), five larvae or adults were collected and pooled for each test. (a) Control experiment without dsRNA application was conducted in two colonies: one was artificially infected with SBV, and the increase in SBV in adult honey bees (Adult bee – SBV) and larvae (Larva – SBV) was evaluated, and the other colony was maintained in a healthy condition without SBV infection and only adult honey bee was used for SBV quantification (Adult bee – no SBV). (b) In another colony, dsRNA was supplied to the hive by spraying 3 days before SBV infection, and the SBV in adult honey bees and larvae was quantified 2, 4, 7, and 10 days after infection.

  • in the figue1, Is there any significant difference between the RNAi + SBV group and the SBV group?

Answer: The average result of three replicated treatments showed that the dsRNA treatment group increase the survival rate to 32.7% compared to the non-treatment group, survival rate of the dsRNA treatment and non-treatment group was 81.5% and 48.8%, respectively. However, p value of comparison between the two groups based on the result of three times experiment was p = 0.08. According to the p value the comparison of two group was not significantly difference. The experiment might need to have more replication to see the significant difference of the two group, this limitation is mentioned in the conclusion Line 374-377.

  • in my opinion, both Figures 3 and 4 illustrate that the amount of SBV is reduced after dsRNA treatment, fig4 could put into supplements.

Answer: The Figure 4 was moved to the supplementary data (Figure S6)

  • there was confused about the quantification of SBV, in the manuscript, KSBV-PCR-F (5′-GACCAAGAAGGGAATCAG-3′) and KSBV-PCR-123R (5′-CATCTTCTTTAGCACCAGTATCCA-3′) was used, if so, the primers in table1 were used for what? In addition, KSBV-PCR-123R should revise to KSBV-PCR-R.

Answer: We designed primer pair and probe for detection of all different genotypes SBV from field collected honey bee samples in the country, the information from our recently published paper is now given in the manuscript (reference number 30), and the Table 1 showing the primer information and PCR condition was removed from the manuscript. However, this primer pair and probe were not efficient for quantitative detection of Korean SBV, the pathogenic genotype in Apis cerana in Korea. Therefore, we designed another primer pair KSBV-PCR-F/KSBV-PCR-123R for quantitative detection of Korean genotype of SBV in Apis cerana colonies. This primer pair target on the specific mutation region of SBV genotype in Korea (Figure S4). The primer name KSBV-PCR-123R was revised to KSBV-PCR-R.

  • the text should be revised with attention because there are a lot of grammatical errors and typos.

Answer: The manuscript was carefully checked and fixed the errors. All the revised positions are marked with track change

Line 129, Table2, T7VP1F, target gene was VP3, could you explain why T7VP1 target VP3?

Answer: The target gene was initially named as VP1 according to the same position in previous publication of Liu et al. (2010) (Liu, X.; Zhang, Y.; Yan, X.; Han, R. Prevention of Chinese sacbrood virus infection in Apis cerana using RNA interference. Curr. Microbiol. 2010, 61, 422–428, doi:10.1007/s00284-010-9633-2), the reference is added in the Table 1. However, in recently published paper (Procházková, M.; Füzik, T.; Škubník, K.; Moravcová, J.; Ubiparip, Z.; PÅ™idal, A.; Plevka, P. Virion structure and genome delivery mechanism of sacbrood honeybee virus. Proc. Natl. Acad. Sci. U. S. A. 2018, 115, 7759-7764, doi:10.1073/pnas.1722018115) the position was identified to be VP3. Therefore, the name of target gene was changed to be VP3, and primers was revised to be T7VP3F in the Table 1.

Line 161,”for 1 m to eliminate ” should revise to “for 1 min to eliminate”;

Answer: The word “ 1 m” was revised to be 1 min in the manuscript.

Reviewer 2 Report

Dear Editor,

This is interesting paper aimed to combat with infection of SBV in managed honey bee on the basis of RNA interference process.

I have some comments and suggestions to improve quality of the paper.

Line 23. Meanwhile - please avoid repeating words at the beginning of the sentence.

Line 32. Please, add belongs to family of Iflaviridae (part of the group within the order Picornavirales).

Line 34.  different functional proteins – replace with nonstructural proteins.

Line 35-36. Please note that the virus mostly affects worker larvae, but can also infect adult honey bees.

Line 38-39. Please, replace with a mutation in the V4 (minor capsid protein gene).

Line 45. Before the next sentence, the author may can list some of the main approaches for combat against the virus

Line 57. The capacity of RNA interference (RNAi) to inhibit – what are the results of the application?

Line 71. sacbrood virus – Sacbrood virus.

Table 1. – The RT-PCR conditions are very strange.

Line 78. The SBV band – how the authors visualized this band?

Line 99. PCR – replace with RT-PCR.

Line 107. Add Republic of Korea in bracket. Also, nothing mention about bacteria in which the vector was cloned.

Line 119. 1 μg of linear DNA template – this is misleading, please correct? Also, linear DNA template should be cDNA.

Line 149-150. May be the author should be state infected artificially reared larvae.

Line 156. SBV used for infection was collected from field SBV-positive samples – I do not understand this. What are field SBV-positive samples?

Line 181. The experiment if each group was repeated three times - over what period of time?

Line 223. Please, replace GFP gene with GFP protein.

Line 301. the cost was lower compared with that of in vitro production – Please, view previous sentence.

Author Response

Reviewer 2

This is interesting paper aimed to combat with infection of SBV in managed honey bee on the basis of RNA interference process. I have some comments and suggestions to improve quality of the paper.

Answer: Thank you for your comments and suggestions to help us improve the quality of the manuscript, we revised the manuscript according to each comment indicated below:

Line 23. Meanwhile - please avoid repeating words at the beginning of the sentence.

Answer: The word “meanwhile” in Line 23 was replaced with “However”.

Line 32. Please, add belongs to family of Iflaviridae (part of the group within the order Picornavirales).

Answer: The phrase “belongs to family of Iflaviridae” was added in the sentence in Line 32-33

Line 34. Different functional proteins – replace with nonstructural proteins.

Answer: The sentence was revised by substituting “nonstructural proteins” for “different functional proteins” in Line 34

Line 35-36. Please note that the virus mostly affects worker larvae, but can also infect adult honey bees.

Answer: The information of SBV infection in adult bee is added in the manuscript in Line 37-39

Line 38-39. Please, replace with a mutation in the V4 (minor capsid protein gene).

Answer: According to the analysis of Procházková et al. (2018) (Procházková, M.; Füzik, T.; Škubník, K.; Moravcová, J.; Ubiparip, Z.; PÅ™idal, A.; Plevka, P. Virion structure and genome delivery mechanism of sacbrood honeybee virus. Proc. Natl. Acad. Sci. U. S. A. 2018, 115, 7759-7764, doi:10.1073/pnas.1722018115) the mutation position located at the newly identified protein named “minor capsid protein”, this position differs from the that of VP4 gene. Therefore, the “minor capsid protein gene” in Line 42-43 is used.

Line 45. Before the next sentence, the author may can list some of the main approaches for combat against the virus

Answer: Some main methods for treatment of SBV disease are listed in the manuscript in Line 46-47.

Line 57. The capacity of RNA interference (RNAi) to inhibit – what are the results of the application?

Answer: The result of small hive beetle and Varroa destructor mite inhibition by RNAi is added in the manuscript between Line 63-65

Line 71. sacbrood virus – Sacbrood virus.

Answer: The name of virus is revised with the capital letter in Line 77.

Table 1. – The RT-PCR conditions are very strange.

Answer: The accuracy of RT-qPCR condition was confirmed, the material and RT-qPCR condition were according to the LiliF™ SBV/KSBV/DWV/BQCV RT-qPCR kit (iNtRON Biotechnology, Inc., Seongnam, Korea). Reverse transcription was done for 30 min at 45℃ followed by 10 min at 95℃ for inactivating reverse transcriptase and pre-denaturation of cDNA, then 40 cycles of two steps real-time PCR. This information was shown in our recently published paper and was cited in the manuscript (reference number 30). Therefore, the primers and PCR condition Table 1 was removed.

Line 78. The SBV band – how the authors visualized this band?

Answer: After purification and ultracentrifugation in the sucrose gradients the large amount of the virus can be seen by naked eyes with a white band located at the position corresponding to the concentration of 40% sucrose. The band was extracted for purification then observation and taking photo under a transmission electron microscope (Figure S1).   

Line 99. PCR – replace with RT-PCR.

Answer: Reverse transcription was firstly carried out using SuperScript III First-Strand Synthesis System for RT-PCR (Thermo Fisher Scientific, Wal-tham, MA), then the cDNA was used for PCR. Only PCR was performed using the cDNA solution in this step. Therefore, the term “PCR” is used in Line 112.  

Line 107. Add Republic of Korea in bracket. Also, nothing mention about bacteria in which the vector was cloned.

Answer: The full name “Republic of Korea” and its abbreviation “ROK” was firstly mentioned in the main text in Line 41. After that the abbreviation form “ROK” is used in all other positions. The recombinant plasmid was transformed into E. coli DH5 (Enzynomics, Daejeon, ROK) for cloning. The information of bacteria is added in the manuscript in Line 120-121.   

Line 119. 1 μg of linear DNA template – this is misleading, please correct? Also, linear DNA template should be cDNA.

Answer: PCR product of VP3 gene was used as DNA template for production of dsRNA. The information is provided in Line 129-130. The term “of linear” was removed from the sentence in Line 135.

Line 149-150. May be the author should be state infected artificially reared larvae.

Answer: The experiment was also conducted to evaluate the safety of dsRNA to the healthy larvae (no SBV infection). Therefore, the term “artificially reared larvae” is used in Line 165.

Line 156. SBV used for infection was collected from field SBV-positive samples – I do not understand this. What are field SBV-positive samples?

Answer: The source of SBV used for infection to the reared larvae is originated from the larval samples that were collected from the natural infected colonies in apiaries. The sentence was reworded to clarify the intended meaning in Line 170-173.  

Line 181. The experiment if each group was repeated three times - over what period of time?

Answer: Three independent trials were done, and the period of each trial was 8 days. The information is added in Line 198.  

Line 223. Please, replace GFP gene with GFP protein.

Answer: The word “gene” was deleted from the sentence in Line 248

Line 301. the cost was lower compared with that of in vitro production – Please, view previous sentence

Answer: The sentence was reworded to clarify the intended meaning in Line 331-334

Reviewer 3 Report

The manuscript is important and of interest due to its attempt to address the important issue of the absence of viable, scalable interventions for viral diseases which threaten honey bee colonies globally. This particular paper is a nice starting point but the field studies, frankly, leave a lot of room for improvement. Most notably they are poorly controlled which severely limits the scope of the claims being made and leave one wondering whether the target chosen by the authors really do scale to the field. Below are specific comments to be addressed:

Points to clarify or address before publication include:

1.     In the interest of full transparency and accessibility I want to strongly encourage the authors to include all raw data (for example the qPCR data) in an accessible format like excel or a .csv file so that readers may reanalyze their samples themselves.

2.     I am not sufficiently convinced by the field data that their dsRNA application successfully treated SBV infection. At the very least there is zero evidence produced that they were able to prevent SBV infection.

                                               i.     Where were the apiaries in comparison to one another? Should/could we reasonably suspect that the ‘control’ apiary would have a similar disease pressure to the test apiaries?

                                             ii.     The authors sampled larvae in pools of 2. How were larvae chosen? Was there a particular instar chosen? Did the authors look for signs of symptomology prior to sampling? If samples are taken in proximity of symptomatic larvae one might expect higher levels of SBV which could skew data.

                                            iii.     Table 4 presents the results of a large scale field trial. There are no control colonies which went on to develop disease/infection so I don’t see how the authors can claim this is evidence of prevention. It seems equally likely it could be that these colonies simply never would have developed infection in any case.

                                            iv.     Table 4: The authors have stated that the colonies treated in 1 week intervals have “less pulled out larvae”. Is this a qualitative measure or did the authors quantify this?

                                              v.     Table 4: What were the rates of colony loss? The authors should clarify if they mean that all symptomatic colonies treated in 4 week intervals died “within 2 months”, for example. Or was it some percentage of these colonies?

                                            vi.     Lines 306-308: “Therefore, during the field application of dsRNA for the disease prevention in infected colonies, the queens were confined for the first 3 weeks”

a.     I’m not sure if this is in reference to the present study or not. Please clarify. If the queens were trapped in dsRNA treated hives were they also trapped in control hives?

                                           vii.      

3.     The ‘Data Analysis’ section of the methods must be rewritten to be clearer. Which analyses were applied to which experiments? And, the statistical tests used to for analyses should be identified in the results section. For example, in paragraph 1 of the results the authors state:

“This was demonstrated by the survival rate (92.3%) of the group that was fed only 215 dsRNA (without SBV) not significantly differing from that (94.6%) of the larvae that only 216 received the feeding solution without SBV (p = 0.66; Figure 1).”

They should indicate in the parentheses which test was used.

4.     Further, they state that infected larvae that received dsRNA and SBV survived at a higher rate (81.5%) relative to those that did not receive dsRNA (48.8%). This is nominally a higher rate but since the authors did not break these numbers down by replicate it’s unclear if this difference is significant. Appropriate statistics should be applied to evaluate the difference.

5.     Why did the authors stop at day 8 for their mortality? Is this normal time to pupation for Apis cerana? Please clarify.

Minor points to clarify or address:

1.     Lines 36-38: The authors indicate that Apis cerana acquired SBV from Apis mellifera. Is there a definitive citation for this claim?

2.     Lines 98-99: More detail regarding cDNA synthesis is needed. How much RNA was used? What was the final concentration of primer (random hexamer or oligo dT)?

3.     Lin 119: What was the final concentration of NTPs?

4.     Line 126 and Figure S3: The gel pretty clearly shows higher molecular weight bands above the dsRNA band. What are these?

5.     Line 138: “2 uL of sample RNA” – did the authors not standardize RNA weight input at all? This could be a significant error when trying to compare viral genome quantities.

6.     To that end – how did the authors calculate “Log10[SBV cDNA copes]/bee” as per figure?

7.     Line 155: I think the authors indicate later in that paragraph that larvae were orally inoculated. Please state that explicitly in this sentence.

8.     Line 192: “..’RT-qPCR, with 7.77 × 105–2.11 × 108 copies of SBV DNA per larva.’.

a.     I believe these numbers are referring to the colonies in table 3? Where are these data? If the authors have quantitative data for these colonies why are they not included here?

9.     Line 193: “…The hives were administered with 193 dsRNA five times, with a 1-week interval.” What were the dates? For posterity it would be helpful to know what time of year these experiments took place as viral infections are highly variable across time and space.

10.  Lines 197-198: “Finally, dsRNA was applied for the prevention and treatment of SBV in large-scale 197 in 269 colonies belonging to 33 different apiaries in ROK, with oral administration of 20 198 mg dsRNA/hive.”

a.     For practical reasons, can the authors state a cost estimate here. Cost per mg of dsRNA and/or per colony would suffice. It’s not clear how this method solves any problems of scalability.

11.  Line 263: It is a major oversight that the authors only chose a single control colony. As far as the authors have indicated the reader has no reason to believe this is representative of control colonies. At best the authors need to identify these limitations in the discussion.

12.  so that the number of larvae in the colony did not increase any further.

13.  Figure 3: Can the authors please replot this with dots for each measurement taken? This will make it more clear on first glance how many data points are used to make these inferences and what the actual variation is in the data.

14.  Figure 4: Are these adult bees being tested or larvae? How many samples are being tested per colony? If more than one, please replot these data so that we can see what the variability is within colonies.

15.  Figure S4: Please provide the alignment in an accessible format like a text file or a fasta file.

Author Response

Reviewer 3

The manuscript is important and of interest due to its attempt to address the important issue of the absence of viable, scalable interventions for viral diseases which threaten honey bee colonies globally. This particular paper is a nice starting point but the field studies, frankly, leave a lot of room for improvement. Most notably they are poorly controlled which severely limits the scope of the claims being made and leave one wondering whether the target chosen by the authors really do scale to the field. Below are specific comments to be addressed:

Answer: Thank you for the helpful comments and suggestions. We agree with the comments of reviewer and revised the manuscript according to each indicated comment and suggestion as bellow:

  1. In the interest of full transparency and accessibility I want to strongly encourage the authors to include all raw data (for example the qPCR data) in an accessible format like excel or a .csv file so that readers may reanalyze their samples themselves.

Answer: The raw data of threshold cycle of SBV detection and quantification was conducted in 2017. The manuscript was written using the data from the final report of the project in 2018. The person who performed the RT-qPCR quit the work the institute and moved to another company for new work, we sent email to ask for the raw data. Unfortunately, during the revising period we have not received the reply.

  1. I am not sufficiently convinced by the field data that their dsRNA application successfully treated SBV infection. At the very least there is zero evidence produced that they were able to prevent SBV infection.
  2. Where were the apiaries in comparison to one another? Should/could we reasonably suspect that the ‘control’ apiary would have a similar disease pressure to the test apiaries?

Answer: In case if all the colonies of dsRNA treatment group and control group are free with SBV the disease pressure might not similar among the two groups. However, in this study the colonies in both treatment and control groups selected for the evaluation were all positive with SBV in the RT-PCR detection. The result of SBV detection and quantification in the colonies before application of dsRNA were shown in the Test 1 in the Figure 4 (it is now Figure S6). Therefore, the selected colonies in this study were all facing with the same risk of SBV disease.

  1. The authors sampled larvae in pools of 2. How were larvae chosen? Was there a particular instar chosen? Did the authors look for signs of symptomology prior to sampling? If samples are taken in proximity of symptomatic larvae one might expect higher levels of SBV which could skew data.

Answer: Pool of 5 larvae or 2 adult bees were sampled from each colony for the quantitative detection of SBV in the dsRNA treatment colonies, it was mentioned in the method section Line 92 and 95. The information of SBV quantification from dsRNA treatment colonies shown in Figure 3 is from the adult bee because during the first three weeks of treatment period the queen were confined, and the larvae were decrease from the second week. Therefore, adult bees were used for the detection and quantification of SBV. The information in Figure 3 is corrected. For the detection of SBV in apiaries the larval samples were used, the 3rd to 5th instar larvae were randomly collected for SBV detection. In the colonies administrated with dsRNA although SBV was detected in RT-PCR, there was no symptom of SBV disease was seen.

iii. Table 4 presents the results of a large-scale field trial. There are no control colonies which went on to develop disease/infection so I don’t see how the authors can claim this is evidence of prevention. It seems equally likely it could be that these colonies simply never would have developed infection in any case.

Answer: After examining the efficiency of SBV prevention in the small-scale study and the high rate of SBV disease in the control colonies the dsRNA was then provided to the beekeepers in the whole country for SBV prevention in 2017, during the administration of dsRNA for the prevention purpose no symptom of disease was observed in these colonies. However, in the large-scale application we did not set the colony for the negative control.

  1. Table 4: The authors have stated that the colonies treated in 1 week intervals have “less pulled out larvae”. Is this a qualitative measure or did the authors quantify this?

Answer: The symptom of SBV disease can be seen in the larvae that were pulled out and deposited in front of the hive (Figure 2b). However, we did not quantify the number of larvae pulled out.

  1. Table 4: What were the rates of colony loss? The authors should clarify if they mean that all symptomatic colonies treated in 4 week intervals died “within 2 months”, for example. Or was it some percentage of these colonies?

Answer: In the colonies with symptom of disease the treatment with different interval can extend the duration of colonies survival. However, finally all the colonies (100%) were died. The information is added in the Table (it is now Table 3).  

  1. Lines 306-308: “Therefore, during the field application of dsRNA for the disease prevention in infected colonies, the queens were confined for the first 3 weeks”. I’m not sure if this is in reference to the present study or not. Please clarify. If the queens were trapped in dsRNA treated hives were they also trapped in control hives?

Answer: In the small-scale treatment of dsRNA the queens were confined in both treated colonies and the control colonies for the first three weeks during the period of dsRNA treatment (for 5 weeks). For the queen trapping to mitigate the development of disease we did not consult the reference. Therefore, no reference was cited here.

  1. The ‘Data Analysis’ section of the methods must be rewritten to be clearer. Which analyses were applied to which experiments? And, the statistical tests used to for analyses should be identified in the results section. For example, in paragraph 1 of the results the authors state:

“This was demonstrated by the survival rate (92.3%) of the group that was fed only 215 dsRNA (without SBV) not significantly differing from that (94.6%) of the larvae that only 216 received the feeding solution without SBV (p = 0.66; Figure 1).” They should indicate in the parentheses which test was used.

Answer: The data analysis part between Line 221-226 was revised by indicating the groups of comparison that corresponding to the result section. The groups were also indicated in parentheses in the result section between Line 235-244.   

  1. Further, they state that infected larvae that received dsRNA and SBV survived at a higher rate (81.5%) relative to those that did not receive dsRNA (48.8%). This is nominally a higher rate but since the authors did not break these numbers down by replicate it’s unclear if this difference is significant. Appropriate statistics should be applied to evaluate the difference.

Answer: Although the survival rate of dsRNA treatment group was higher than in control group (32.7%). However, the significant different was not seen according to the p value of comparison (p = 0.08). The result could be due to the data from low number of replication (three times). The result is added in the result section in Line 237-238. The limitation was also mentioned in the conclusion section Line 375-376.

  1. Why did the authors stop at day 8 for their mortality? Is this normal time to pupation for Apis cerana? Please clarify.

Answer: According to the previous publication about rearing larvae of Apis cerana in vitro (Toan TV, Lee ML, Sim HS, Kim HK, Byuon GH, Choi YS. 2014. Initial Results of Rearing Honey bee Apis cerana in vitro. Journal of Apiculture 29(3): 193~197) the time for pupation of A. cerana larvae is from the 8th day. We used the 3rd instar larvae for the experiment, and stop the observation at the 8th day, to make sure that the time is enough to see the mortality of the larvae because the survival larvae were transferred to the pupal stage that can be easily identified as showing in the Figure below:

Figure. SBV infected larvae reared in vitro during the period of dsRNA. The 3rd instar larvae were used for the experiment. The photos were taken at the 2nd (A), 4rd (B), 6th (C), and the 8th day during the period of treatment. At the 8th day of the treatment corresponding to the 11th of the life time of the brood. The survival larvae transferred to the pupal stage, the dead larvae can be identified by failing pupation and change the color to be brown (D).

Minor points to clarify or address:

  1. Lines 36-38: The authors indicate that Apis cerana acquired SBV from Apis mellifera. Is there a definitive citation for this claim?

Answer: The sentence was corrected in Line 39

  1. Lines 98-99: More detail regarding cDNA synthesis is needed. How much RNA was used? What was the final concentration of primer (random hexamer or oligo dT)?

Answer: The procedure of cDNA synthesis is added in the manuscript between Line 105-112, 4 µL of total RNA and 5 µL of reverse primer (T7VP3R; 10 pmol/µL; Table 1) were used for the synthesis.

  1. Line 119: What was the final concentration of NTPs?

Answer: 2.5 µL of each NTP (50 nmol/µL) was used for the synthesis of dsRNA. The information is added in Line 134.  

  1. Line 126 and Figure S3: The gel pretty clearly shows higher molecular weight bands above the dsRNA band. What are these?

Answer: The Figure S3 showing the band of dsRNA that was produced with large quantity and low price (10USD for 20mg) by the partner company Genolution, Inc. (Seoul, ROK). And then, in my laboratory the sequences of target dsRNA were confirmed as the same as targeted VP3 region.

  1. Line 138: “2 uL of sample RNA” – did the authors not standardize RNA weight input at all? This could be a significant error when trying to compare viral genome quantities.

Answer: To avoid the variation among the samples we used the same volume of RNA solution (2 µL) for each reaction from all samples, the same volume of RNA solution extracted from each sample was 50 µL. However, we did not measure the concentration of RNA in the solution used for RT-qPCR.

  1. To that end – how did the authors calculate “Log10[SBV cDNA copes]/bee” as per figure?

Answer: The copy number of initial DNA template in each RT-qPCR reaction was firstly calculated based on the cycle threshold (CT) of detection using the standard linear regression created from standard recombinant DNA (Figure S5). This initial DNA copy equal to the copy number of SBV cDNA in 2 µL RNA solution used for each RT-qPCR reaction. The number was divided by 2 to calculate the number of cDNA/µL, and then multiplied by 50 µL (total volume of extracted RNA from each sample) to calculate the total number of SBV cDNA could have in the RNA solution extracted from each sample. Finally, the total cDNA number was divided by 2 (adult bees) or 5 (larvae) to have the number of SBV cDNA per bee, then log10 of the cDNA per bee and presenting it on the figure.

  1. Line 155: I think the authors indicate later in that paragraph that larvae were orally inoculated. Please state that explicitly in this sentence.

Answer: The artificially reared larvae were used for evaluation of SBV infection and dsRNA treatment. To do that, firstly we prepared the SBV virus by extraction and purification from the infected samples from apiaries, then mix the virus with feeding solution to feed the reared larvae. The sentence is revised in Line 171-173.

  1. Line 192: “..’RT-qPCR, with 7.77 × 105–2.11 × 108 copies of SBV DNA per larva.’.
  2. I believe these numbers are referring to the colonies in table 3? Where are these data? If the authors have quantitative data for these colonies why are they not included here?

Answer: Before supplying dsRNA to the colonies we did quantitative detection of SBV to confirm the presence of SBV in the colonies. The result is shown in Figure 3 and the Test 1 in Figure S6. However, the result belongs to the result section. Therefore, we did not indicate it in the material and methods section. The information in Line 209 was removed.

  1. Line 193: “…The hives were administered with 193 dsRNA five times, with a 1-week interval.” What were the dates? For posterity it would be helpful to know what time of year these experiments took place as viral infections are highly variable across time and space.

Answer: The experiment of dsRNA treatment was done during summer season from May to June. In this season the disease highly spread in the apiary in South Korea. the information is added in the manuscript in Line 211.

  1. Lines 197-198: “Finally, dsRNA was applied for the prevention and treatment of SBV in large-scale in 269 colonies belonging to 33 different apiaries in ROK, with oral administration of 20 mg dsRNA/hive.”
  2. For practical reasons, can the authors state a cost estimate here. Cost per mg of dsRNA and/or per colony would suffice. It’s not clear how this method solves any problems of scalability.

Answer: After evaluating the high efficiency of the dsRNA for SBV prevention we cooperate with the company Genolution, Inc. (Seoul, ROK) to produce the large quantity for the large-scale application, the cost for each 20 mg dsRNA was 10USD. The information of dsRNA and the price of the dsRNA can be verified by the company. The information is given in Line 217-218. However, the detail method for large scale production was not provided by the company due to the economic confidentiality.

  1. Line 263: It is a major oversight that the authors only chose a single control colony. As far as the authors have indicated the reader has no reason to believe this is representative of control colonies. At best the authors need to identify these limitations in the discussion.

Answer: The dsRNA is useful and widely used in the country for SBV prevention and treatment. This manuscript we report our initial evaluation of the use of dsRNA. However, the limitation of experiment design is remained. Therefore, we added sentences in the conclusion section between Line 375-377 to state the limitation of our study.

  1. so that the number of larvae in the colony did not increase any further.

Answer: In the first application of dsRNA in the small scale of colonies we confined the Queen for the first three weeks to mitigate the number of infected larvae. After that the queen was released. In the large scale we did not confined the queen, but the colonies received dsRNA all showed no symptom of SBV disease during the period of experiment. The sentence in Line 340 was removed from the manuscript.

  1. Figure 3: Can the authors please replot this with dots for each measurement taken? This will make it more clear on first glance how many data points are used to make these inferences and what the actual variation is in the data.

Answer: The Figure 3 was replotted with dots format. For SBV detection pool of 5 larvae or 2 adult bees from each colony was used for each test (mentioned in Line 92 and 95). Therefore, the information in the Figure 3 was corrected.

  1. Figure 4: Are these adult bees being tested or larvae? How many samples are being tested per colony? If more than one, please replot these data so that we can see what the variability is within colonies.

Answer: The Figure 4 and Figure 3 showed the same data of SBV detection and quantification from the 22 colonies. However, in the Figure 4 the results were shown in detail from all colonies in each apiary in 5 tests with 1-week interval. The adult sample (2 bees from each colony) were used for the test. This figure is now moved to supplementary data Figure S6.  

  1. Figure S4: Please provide the alignment in an accessible format like a text file or a fasta file.

Answer: The alignment file used for primer design for specific detection of Korean SBV is provided in a Fasta file.
